# INFER: IMPLICIT NEURAL FEATURES FOR EXPOSING REALISM

## ABSTRACT

Deepfakes pose a significant threat to the authenticity of digital media, with current detection methods often falling short in generalizing to unseen manipulations. *INFER* is the first deepfake detection framework that leverages Implicit Neural Representations (INRs), marking a new direction in representation learning for forensic analysis. We combine high-level semantic priors from Contrastive Language–Image Pre-training (CLIP) with spatially detailed, frequency-sensitive features from INR-derived heatmaps. While CLIP captures global context grounded in natural image statistics, INR heatmaps expose subtle structural inconsistencies often overlooked by conventional detectors. Crucially, their fusion transforms the feature space in a way that enhances class separability—effectively re-encoding both spatial artifacts and semantic inconsistencies into a more discriminative representation. This complementary integration leads to more robust detection, especially under challenging distribution shifts and unseen forgery types. Extensive experiments on standard deepfake benchmarks demonstrate that our method outperforms existing approaches by a clear margin, highlighting its strong generalization, robustness, and practical utility.

## 1 INTRODUCTION

With the rapid progress of deep learning, it has become easier than ever to generate highly realistic synthetic media, including images, videos, and audio. One of the most widely known and debated results of this technology is deepfakes, which is artificial content that is designed to closely mimic real-world media. Today, a deepfake is typically defined as any image, video, or audio clip that has been generated or modified using deep learning methods, often to deceive viewers or mislead them into believing the content is authentic. The term deepfake comes from a combination of deep, referring to deep learning, and fake, indicating that the content is not genuine. Although early attempts to alter video content go back to the 1990s, such as the Video Rewrite system (1997), which altered a person's lip movements in video to match different audio (Norman, 2025); these methods did not involve deep neural networks. The modern concept of deepfakes only became possible with the rise of powerful deep learning models. In particular, Generative Adversarial Networks (GANs) (Singh et al., 2020; Alqahtani et al., 2021) played a major role in creating realistic synthetic faces and videos. More recently, diffusion models (Croitoru et al., 2023) have made it possible to generate even more seamless and photo-realistic content that is difficult to distinguish from real media (Bhattacharyya et al., 2024; AV et al., 2024). As deepfake technology becomes increasingly advanced, and widely accessible (Lanzino et al., 2024), the creation of synthetic media is accelerating at a rapid pace. Recent estimates suggest that thousands of deepfakes are now being generated daily, with applications ranging from entertainment and satire to more harmful uses such as misinformation campaigns, identity theft, and financial fraud (Hancock & Bailenson, 2021; de Rancourt-Raymond & Smaili, 2023; Gilbert & Gong, 2024). These growing risks have sparked widespread concern around media authenticity and digital trust.

In response to the growing threat of deepfakes, researchers have turned to the same technology that enabled their creation, which is deep learning, to develop effective detection methods. Broadly, deepfake detection techniques fall into two main categories: image-based and video-based approaches (Heidari et al., 2024). Image-based methods focus on analyzing individual frames to identify visual artifacts or inconsistencies, and are often simpler and faster to train (Altaei et al., 2023; Raza et al., 2022; Frank et al., 2020). In contrast, video-based methods aim to capture temporal inconsistencies

across frames, such as unnatural facial expressions, blinking patterns, or head movements, but typically require more complex models and greater computational resources (Suratkar & Kazi, 2023; Yu et al., 2021; Kaur et al., 2024).

While a wide range of deepfake detection methods have been proposed, a persistent challenge remains: generalization to unseen manipulations and datasets. Many models perform well on specific benchmarks but struggle when faced with new deepfake generation techniques or distribution shifts in real-world data. This raises a critical question: *What types of representations can lead to better class separation and more robust detection than traditional approaches?* One promising direction involves the use of features derived from Contrastive Language–Image Pre-training (CLIP) (Radford et al., 2021). Recent studies have shown that CLIP features, which encode high-level semantic and visual information, offer improved class separability compared to existing methodologies (Ojha et al., 2023). Building on this, further work has demonstrated that applying wavelet decomposition to CLIP-derived features can capture localized frequency components, leading to enhanced detection performance (Baru et al., 2024).

These insights strongly suggest that combining semantic-rich embeddings with frequency-aware representations may offer a promising path toward more generalizable deepfake detection. Motivated by this, we seek an alternative representation, that can be combined with CLIP embeddings, which not only captures frequency characteristics but also retains spatial context, enabling the model to reason about where and how manipulations occur within an image. While many decomposition methods exist, we observe that Implicit Neural Representations (INRs) (Sitzmann et al., 2020) offer a unique formulation. They model images as continuous functions over spatial coordinates, implicitly encoding both fine-grained structure and frequency content within their network parameters. In doing so, the layer-wise activations of INRs naturally act as a form of spectral decomposition (Benbarka et al., 2022), revealing localized frequency responses across the image. Unlike traditional CNNs that operate on fixed grids, INRs provide a flexible and expressive representation that has recently shown promise across various signal domains, including images, audio, and video (Sitzmann et al., 2020; Ramasinghe & Lucey, 2022; Saragadam et al., 2023). This makes them particularly well-suited for capturing the subtle artifacts introduced by generative manipulations. By leveraging the representational power of INRs, we aim to build a more robust and manipulation-sensitive feature space that complements high-level semantic cues and improves generalization to unseen deepfake types. To the best of our knowledge, *this work is the first to explore the use of INRs for deepfake detection, leveraging their spatial-frequency sensitivity to identify manipulation artifacts.*

## 2 RELATED WORKS

### 2.1 DEEPFAKES

Deepfake detection has become a widely studied domain due to the rise of powerful generative models. Early methods (Afchar et al., 2018; Stehouwer et al., 2019; Li & Lyu, 2018) employ a feature encoder followed by a binary classifier to predict manipulated content. XceptionNet (Chollet, 2017) is based on depthwise separable convolutions with residual connections. Similarly, CapsuleNet (Nguyen et al., 2019) better captures spatial hierarchies in manipulated media. However, these approaches were prone to overfitting and exhibited poor generalization to unseen data. The current deepfake detection landscape can be categorized along two major axes: frame-level vs. video-level detection methods and spatial domain vs. frequency domain methods. Frame-level methods (Shi et al., 2025; Huang et al., 2023; Larue et al., 2023) analyze individual frames for manipulation without considering temporal consistency. Video-level methods (Wang et al., 2023; Xu et al., 2023; Haliassos et al., 2022) leverage temporal information across frames to enhance robustness. When it comes to spatial domain approaches (Ni et al., 2022; Zhao et al., 2022), they detect inconsistencies at the pixel level. On the other hand, frequency domain approaches (Li et al., 2024; Tan et al., 2024; Jeong et al., 2022) focus on spectral artifacts introduced during manipulation. Recently, several works such as LSDA (Yan et al., 2024) and SBI (Shiohara & Yamasaki, 2022) have proposed dataset augmentation strategies to increase dataset size with high-quality synthetic samples, which has been shown to improve model performance. In contrast, we deliberately avoid using any augmentations in order to highlight the efficacy of INRs in implicitly capturing subtle manipulation artifacts in spatial-spectral domains. Consequently, for a fair comparison, we exclude baselines that employ dataset augmentation. (Ojha et al., 2023) shows the advantage of using semantic CLIP features for

deepfake detection. Wavelet-CLIP (Baru et al., 2024) appends it with additional frequency features obtained using wavelet transform to further improve performance. In our approach, we leverage the superior spatial-spectral decomposition capability of INRs, combined with the semantic richness of CLIP features. Our work falls under the frame-level detection category and utilizes spatial-spectral information derived from INRs to improve deepfake detection performance without the aid of data augmentations.

## 2.2 IMPLICIT NEURAL REPRESENTATIONS

INRs are neural networks that model continuous signals (e.g., images, audio, video) by mapping input coordinates to signal values (Sitzmann et al., 2020). Unlike discrete representations, they embed the signal directly in network parameters, allowing smooth interpolation, compact storage, and high-resolution reconstruction (Saragadam et al., 2023). This makes them well-suited for capturing fine-grained structures and spectral properties. A key factor in their expressiveness is the activation function. Standard choices such as ReLU, Sigmoid, and Tanh fail to preserve high-frequency details. Positional embeddings (PEs) were introduced to inject high-frequency information (Tancik et al., 2020), but often generalize poorly to unseen coordinates. Sinusoidal activations with tailored initialization (Sitzmann et al., 2020) addressed this limitation, while more recent spatial–spectral compact activations improve generalization and relax initialization constraints (Ramasinghe & Lucey, 2022; Saragadam et al., 2023).

The most prominent use of INRs has been in Neural Radiance Fields (NeRFs) (Gao et al., 2022), where they model 3D scenes as continuous volumetric functions for photorealistic view synthesis. Beyond NeRFs, INRs have been applied to image and video restoration tasks such as super-resolution (Aiyetigbo et al., 2025), denoising (Saragadam et al., 2023; Xu et al., 2022; Kim et al., 2022), deblurring (Lehtonen, 2024), inpainting (Xu & Jiao, 2023), and compression of visual data (Strümpler et al., 2022). They have also been explored in medical imaging (Molaei et al., 2023), audio waveform modeling (Sitzmann et al., 2020), and hyperspectral imaging (Chen et al., 2023; Zhang et al., 2022). These applications highlight the versatility of INRs as compact and expressive representations. However, none have investigated their potential for deepfake detection. Our work is the first to explore this direction, showing that INR-derived activations provide a powerful and discriminative modality for identifying subtle manipulations in visual media.

## 3 METHODOLOGY

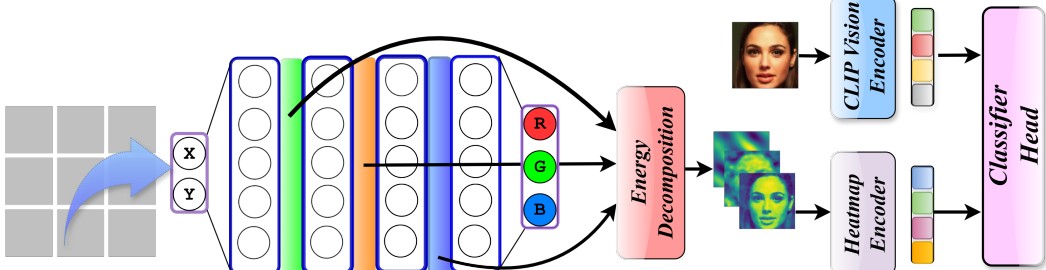

Figure 1: **Overview of the *INFER* Pipeline:** *INFER* begins by associating a spatial coordinate grid with each input image, which is then overfitted using a carefully designed INR. Internal activations from each INR layer are extracted and decomposed using PCA to isolate dominant energy directions. The resulting PCA-based heatmaps are stacked along the batch dimension and processed through a dedicated Heatmap Encoder. In parallel, the RGB image is passed through a CLIP ViT-L/14 encoder to obtain a global semantic embedding. Finally, the INR-derived and CLIP-derived features are concatenated and fed into a classification head for deepfake detection.

## 3.1 DATASET PREPARATION

To build a robust dataset for training and evaluation, we follow a systematic preprocessing pipeline comprising frame extraction, face detection, and alignment. We begin by extracting frames from

each video, followed by face detection using the RetinaFace (Deng et al., 2020) detector. Detected faces are then cropped based on the bounding boxes and aligned using five facial landmark keypoints. The alignment is performed via a *warp and affine* transformation to standardize the facial geometry across samples. All faces are resized following this alignment process. *INFER* is trained on c23 version of the FaceForensics++ (FF++) dataset (Rossler et al., 2019), which simulates realistic video compression artifacts.When it comes to the number of frames, we extract 10 frames per fake video and 40 frames per real video to curate the training set. This sampling strategy ensures a balanced real-to-fake ratio, which helps minimize class bias during training. A critical goal in deepfake detection is to ensure generalization across unseen forgery types. To assess this, we evaluate the trained model on four out-of-distribution (OOD) benchmarks: **Celeb-DF v1** (CDF$_{v1}$) (Li et al., 2020b), **Celeb-DF v2** (CDF$_{v2}$) (Li et al., 2019), **FaceShifter** (FSh) (Rossler et al., 2019), and the **Deep Fake Detection** (DFD) (Rossler et al., 2019) dataset.

## 3.2 Improving deepfake detection via modality fusion

CLIP embeddings have already shown strong performance in deepfake detection (Baru et al., 2024) as it excels in capturing high-level semantic cues such as identity, expression consistency, and scene realism (Asperti et al., 2025). Using a pretrained ViT-L/14 encoder, we extract a global semantic embedding $\mathbf{c} \in \mathbb{R}^{768}$ by feeding in the input image $I$. These features provide robust scene-wide context; however, they may lack explicit spatial and spectral structure.

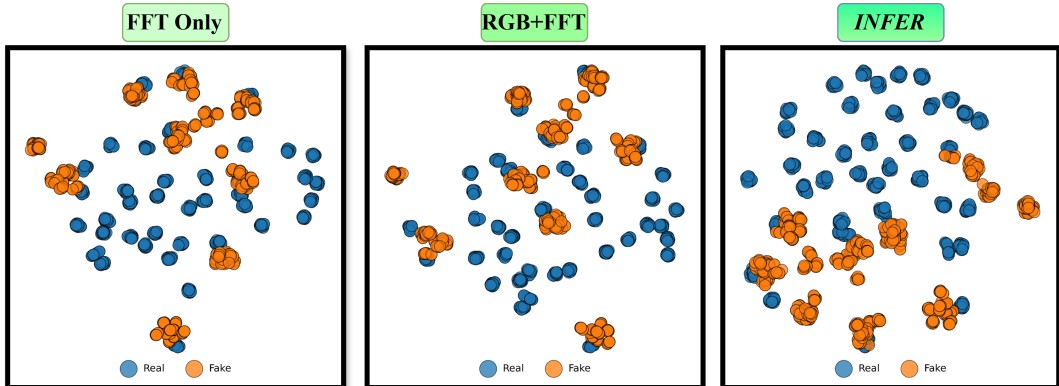

Figure 2: **t-SNE visualization of feature embeddings from the CDF$_{v1}$ dataset using different input modalities**: A clear progression in class separability is observed: FFT-based features show moderate entanglement between real and fake samples, while combining RGB+FFT yields modest improvement by integrating spatial cues. In contrast, *INFER*-derived features exhibit well-defined, compact clusters with a pronounced margin between classes. This suggests that the spatial–spectral decomposition provided by INR heatmaps restructures the feature space in a way that enhances the separability making analogies to the effect of a kernel transformation in classical machine learning

To address this limitation, we explored whether fusing CLIP embeddings with additional modalities could yield improved separability. Specifically, we combined CLIP features with the RGB image and its FFT-based frequency representation (Heckbert, 1995) to inject complementary spatial or spectral information (see Section 4.3 for detailed explanation). However, as seen in both Figure 2 (see the first two figures) and Table 3, even though these conventional representations offer some separation in feature space, greater class separability can be achieved through a further transformation on the feature space. Specifically, the first figure of Figure 2 shows that a degree of separation exists when using FFT. However, the second figure further suggests that combining both FFT and RGB transforms the feature space in a way that enhances class separation even more. This behavior is also reflected in the AUC values reported in Table 3. These observations motivate the idea that modality fusion along with CLIP embeddings can improve class separability, but they also raise the question: which modality can further transform the data to enhance this separation? This motivates the need for a new representation that should ideally include both spatial and spectral features while encoding the required discriminative features. To this end, we explore the possibility of using INRs to derive such features in a multiscale and interpretable manner. The following sections demonstrate on how INRs

can be leveraged alongside CLIP embeddings to improve deepfake detection through enhanced class separability.

### 3.3 FORMULATION OF AN INR

An INR defines a continuous function that maps spatial coordinates $\mathbf{x} \in \mathbb{R}^2$ to RGB values $s(\mathbf{x}) \in \mathbb{R}^3$. This function is typically implemented as a fully connected neural network $f_\theta : \mathbb{R}^2 \to \mathbb{R}^3$, where $\theta$ represents the learnable parameters. Unlike conventional representations (Rabbani & Jones, 1991) that store an image as a discrete grid of pixels, the INR encodes the image in its weights, allowing continuous evaluation at any spatial location. Given a 2D spatial coordinate $\mathbf{x} \in \Omega \subset \mathbb{R}^2$, the network predicts RGB values $\hat{s}(\mathbf{x}) \in \mathbb{R}^3$ through the following layer-wise activations

$$\mathbf{h}_0 = \mathbf{x}, \quad \mathbf{h}_\ell = \phi(\mathbf{W}_\ell \mathbf{h}_{\ell-1} + \mathbf{b}_\ell), \quad \ell = 1, \ldots, L-1, \quad \hat{s}(\mathbf{x}) = \mathbf{W}_L \mathbf{h}_{L-1} + \mathbf{b}_L$$

where $\phi(\cdot)$ is a nonlinear activation (e.g., Sinusoid, Gaussian), and $\mathbf{W}_\ell$, $\mathbf{b}_\ell$ are learnable weights and biases respectively. The network is trained to minimize the MSE loss given by $\mathcal{L}_{\text{recon}} = \frac{1}{|\Omega|} \sum_{\mathbf{x} \in \Omega} \|f_\theta(\mathbf{x}) - s(\mathbf{x})\|_2^2$, where $\Omega$ denotes the set of spatial coordinates in the image domain and $|\Omega| = H \times W$, the $H$ and $W$ represent height and width of the image respectively.

### 3.4 HOW CAN WE DEPLOY INRs FOR DEEPFAKE DETECTION?

#### 3.4.1 LIMITATIONS OF NAÏVE USAGE

A natural question is how INRs can be leveraged for deepfake detection. By design, an INR defines a continuous mapping from spatial coordinates to signal values, offering a compact and differentiable representation of content (Sitzmann et al., 2020). At first glance, this structure seems useful only for reconstruction, with the reconstructed signal then fed to a classifier—essentially no different from using the image itself. This ignores the internal representations unique to INRs. A more promising idea is to treat the INR's learned weights as discriminative features (Malherbe, 2024), but this is computationally demanding. An INR with $L$ fully connected layers of hidden width $d_h$ has about $(d_h^2 + d_h)(L - 2) + 5d_h + 3$ parameters. In practice, reconstructing face images with low error requires at least three hidden layers of width 64, already yielding thousands of parameters. Directly using these weights for classification is therefore expensive and impractical at scale.

#### 3.4.2 SPECTRAL BIAS AND REPRESENTATION DYNAMICS

The challenges noted above motivate the need for more efficient and informative INR representations, especially those unique to INRs yet compact and suitable for downstream tasks. One such direction is to explore structural patterns or emergent behaviors within the weight space. A key insight from the INR literature is *spectral bias* (Rahaman et al., 2019; Yüce et al., 2022), where lower-frequency components of the signal are learned earlier during optimization, while higher frequencies emerge later. Despite its empirical support, there is no definitive theory specifying the number of epochs required to learn each frequency band. Furthermore, as each image, whether real or manipulated, follows its own optimization trajectory, designing a universal schedule or analytical tool to probe weight space remains a challenging open problem.

#### 3.4.3 THE PATHWAY OF A COORDINATE THROUGH THE INR

This challenge can be addressed by analyzing how a spatial coordinate propagates through the layers of an INR, together with the well-known phenomenon of spectral bias. Once trained to minimize reconstruction error, an INR no longer stores an image as pixel values but implicitly encodes it in the network parameters $\theta$ of a function $f_\theta : \mathbb{R}^2 \to \mathbb{R}^3$. Given a coordinate $\mathbf{x} = (x, y)$, the network outputs its RGB value $s(\mathbf{x})$, thereby capturing both spatial layout and frequency content through its parameters (Roddenberry et al., 2023). For each input location, the coordinate is transformed across $L$ layers, producing activations $\{\mathbf{h}_\ell(\mathbf{x})\}_{\ell=1}^{L-1}$ that form a coordinate-conditioned representation path. Each step can be written as $\mathbf{h}_\ell = T_\ell(\mathbf{h}_{\ell-1})$, where $T_\ell$ is the learned mapping at layer $\ell$.

This layered refinement is reminiscent of classical signal decompositions such as wavelets (Zhang, 2019) or multiresolution pyramids (Goutsias & Heijmans, 2000). Unlike handcrafted bases that separate spatial and frequency domains, INRs inherently couple both due to their coordinate-based

formulation. As a result, early layers capture coarse, global structures (low frequencies), while deeper layers encode fine, localized variations (high frequencies), reflecting the spectral bias of neural networks.

### 3.4.4 EXTRACTING INTERPRETABLE FEATURES FROM INR LAYERS.

We begin by examining the internal activations $\mathbf{h}_\ell(\mathbf{x}) \in \mathbb{R}^{d_\ell}$ at each layer $\ell \in \{1, \ldots, L-1\}$ and spatial coordinate $\mathbf{x} \in \Omega \subset \mathbb{R}^2$. These activations form tensors of size $H \times W \times d_\ell$. While these feature maps encode rich information, they are high-dimensional, difficult to interpret, and infeasible to directly use in downstream classification due to memory constraints. To obtain a compact yet informative representation, we seek a transformation that reduces each activation vector to a scalar, while preserving the most structurally meaningful content for deepfake detection. From a signal processing perspective, this corresponds to emphasizing high-energy components—regions where the network's response is most active and discriminative. As an initial step, we explored the $L_2$ norm of the activation vectors. Although smooth and easy to compute, these maps were often dominated by magnitude rather than structure, leading to limited interpretability and poor spatial localization *(See Supplementary material)*. To address this, we adopt a simple, non-learnable alternative that extracts the dominant energy component of each layer's response. Specifically, we use Principal Component Analysis (PCA) to identify the most expressive direction in the activation space. Projecting each feature vector $\mathbf{h}_\ell(\mathbf{x})$ onto this direction yields a scalar heatmap that summarizes the layer's internal representation at each location. The sequence of PCA-derived scalar maps $\{A_\ell(\mathbf{x})\}_{\ell=1}^{L-1}$ forms a structured representation that captures how an INR distributes signal content across layers. We interpret this set as an approximate multiscale decomposition: $I(x, y) \mapsto \mathbf{a}(x, y) := [A_1(x, y), \ldots, A_{L-1}(x, y)] \in \mathbb{R}^{L-1}$.

### 3.4.5 DISCRIMINATIVE NATURE OF THE MULTISCALE DECOMPOSITION

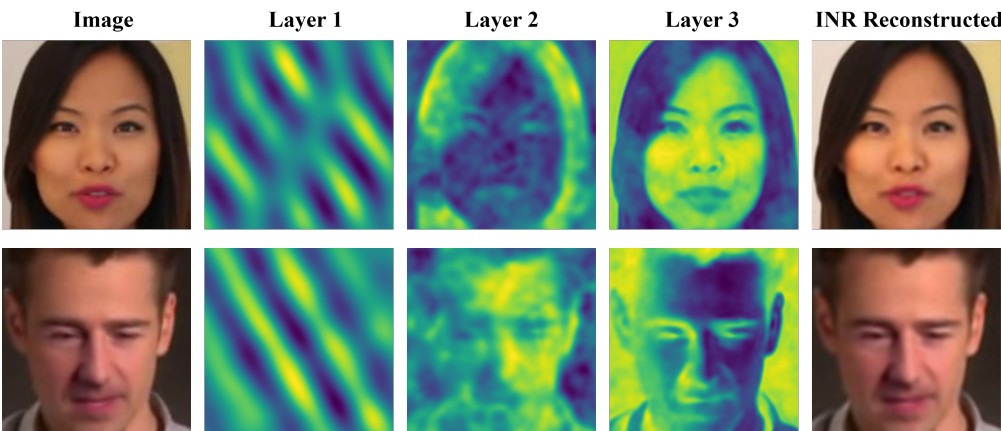

| Image | Layer 1 | Layer 2 | Layer 3 | INR Reconstructed |

Figure 3: **Despite producing visually faithful reconstructions for both real and fake images (last column), INRs exhibit markedly different internal dynamics across layers.** This visualization highlights a key insight about implicit representations: models can generate perceptually accurate outputs while following fundamentally different internal pathways. By projecting layer activations via PCA, these hidden trajectories are revealed—showing that although the output may conceal manipulation, the network's internal structure does not.

Section 3.4.5 presents two examples from the CDF$_{v2}$ dataset: the top row corresponds to a real image, and the bottom row to a deepfake. Each row visualizes the spatial–spectral multiscale decomposition obtained from the INR's internal activations across layers. The final column shows the image reconstructed by the INR, which appears visually similar in both cases despite notable differences in their internal representations. While the quantitative results demonstrate that *INFER* significantly improves deepfake detection across datasets (See Section 4), the proposed decomposition also reveals subtle structural discrepancies, particularly mid-to-deep layers—that are not easily observable in the RGB image or FFT maps. These visual differences provide a glimpse into the discriminative nature

of INR-derived representations, though additional non-visible cues encoded in the internal activations may also contribute to the classifier's decision-making process.

In **Layer 1**, both real and fake activations exhibit wave-like patterns with visually high-frequency textures, which may arise due to the deployed sinusoidal activation function in the INR. Despite their appearance, these early activations primarily capture low-level spatial variations and lack semantic distinction, making them visually similar across real and fake images.

By **Layer 2**, the activations begin to reflect mid-level facial structure. For the real image (top), the representation becomes more coherent where it highlights eyes, nose, and mouth regions with smoother transitions. In contrast, the fake image (bottom) shows irregular, noisy responses lacking semantic consistency. This instability suggests the INR struggles to encode manipulated features cleanly at mid-to-deep levels.

In **Layer 3**, the differences become more pronounced. The real image produces well-aligned, semantically interpretable activations that faithfully reconstruct identity features, whereas the fake image exhibits distorted contours and exaggerated edge responses—visual evidence of manipulation artifacts that become amplified through the INR's encoding process.

Even though the final INR reconstructions (rightmost column) appear visually similar, the internal activations reveal a clear distinction in representation quality.

### 3.5 FUSING SEMANTIC AND MULTISCALE REPRESENTATIONS

To extract robust and discriminative features from the PCA-projected INR heatmaps, we design a compact convolutional encoder tailored to the spatial–spectral nature of these representations. INR-derived heatmaps encode multiscale structural information across layers but can also exhibit smooth gradients and locally diffuse patterns due to the continuity and frequency sensitivity inherent in the INR formulation. Capturing useful cues from such signals requires an architecture that is both spatially aware and resistant to low-frequency redundancy.

We employ stacked $3 \times 3$ convolutional layers to effectively capture local spatial correlations while preserving translational structure. Each convolution is followed by Batch Normalization to stabilize learning and reduce internal covariate shift, and a GELU activation to introduce smooth, non-linear transformations that preserve gradient flow while enhancing expressive capacity. To reduce spatial resolution while retaining global context, we apply an `AdaptiveAvgPool2d` operation that maps the feature maps to a fixed $4 \times 4$ resolution, independent of the input size. This is followed by a fully connected projection and Layer Normalization to produce a compact, fixed-dimensional feature embedding. The heatmap encoder serves as an effective counterpart to the CLIP encoder by transforming localized INR-derived activations into a structured, learnable form. The final CLIP feature and heatmap encoder output are concatenated and passed through a classifier head composed of three fully connected layers with a hidden dimension of 256. This classification module is trained end-to-end using cross-entropy loss to discriminate between real and fake inputs. A visual summary of the entire *INFER* pipeline is shown in Figure 1.

## 4 EXPERIMENTS

### 4.1 EXPERIMENTAL SETUP

To validate the effectiveness of *INFER*, we conduct experiments across multiple deepfake datasets. The training set includes videos generated with four popular manipulation techniques—**Deepfakes**, **Face2Face**, **FaceSwap**, and **NeuralTextures**—covering diverse manipulation styles. The evaluation datasets, detailed in Section 3.1, are distinct in both manipulation technique and visual domain, enabling a rigorous test of generalization. Performance is measured using the Area Under the Curve (AUC), and results for state-of-the-art (SOTA) methods are taken from their respective papers or (Baru et al., 2024).

Table 1 summarizes the performance of the proposed *INFER* compared to existing SOTA methods across four widely-used OOD deepfake detection benchmarks ("–" indicates results not reported in prior works). As evident from the results, *INFER* consistently achieves superior AUC scores, demonstrating strong generalization capability even under distribution shift. For the Celeb-DF family

of datasets, CDF$_{v1}$ and CDF$_{v2}$, *INFER* attains AUC scores of 0.863 and 0.819, respectively. On CDF$_{v1}$, it outperforms the best prior method, SRM (0.792), by a relative margin of **8.22%**. On CDF$_{v2}$, it surpasses the best-performing CLIP-based method, which is Wavelet-CLIP (0.759), by **7.32%**. Notably, when compared against plain CLIP (0.743), the improvement is over **9.28%**, validating the complementary nature of the INR-derived modality. On the FSh dataset, *INFER* achieves an AUC of 0.747, outperforming Wavelet-CLIP (0.732) by a relative margin of **2.00%**. For the DFD dataset, both the F-G method and the proposed *INFER* achieve the same AUC score. It can be stated that, *INFER* delivers consistently strong performance across all benchmarks without requiring dataset-specific tuning or modality customization.

| Model | Venue | CDF$_{v1}$ | CDF$_{v2}$ | FSh | DFD | Avg. |
|---|---|---|---|---|---|---|
| *General SOTA Methods* | | | | | | |
| MesoNet (Afchar et al., 2018) | WIFS-18 | 0.735 | 0.609 | 0.566 | 0.548 | 0.615 |
| MesoInception (Afchar et al., 2018) | WIFS-18 | 0.736 | 0.696 | 0.643 | 0.607 | 0.671 |
| EfficientNet (Tan & Le, 2019) | ICML-19 | 0.790 | 0.748 | 0.616 | 0.815 | 0.742 |
| Xception (Chollet, 2017) | ICCV-19 | 0.779 | 0.736 | 0.624 | 0.816 | 0.739 |
| Capsule (Nguyen et al., 2019) | ICASSP-19 | 0.790 | 0.747 | 0.646 | 0.684 | 0.717 |
| DSP-FWA (Li & Lyu, 2018) | CVPR-19 | 0.789 | 0.668 | 0.555 | 0.740 | 0.688 |
| CNN-Aug (Wang et al., 2020) | CVPR-20 | 0.742 | 0.702 | 0.598 | 0.646 | 0.672 |
| FaceX-ray (Li et al., 2020a) | CVPR-20 | 0.709 | 0.678 | 0.655 | 0.766 | 0.702 |
| FFD (Dang et al., 2020) | CVPR-20 | 0.784 | 0.744 | 0.605 | 0.802 | 0.734 |
| F$^3$-Net (Qian et al., 2020) | ECCV-20 | 0.776 | 0.735 | 0.591 | 0.798 | 0.725 |
| SRM (Luo et al., 2021) | CVPR-21 | 0.792 | 0.755 | 0.601 | 0.812 | 0.740 |
| CORE (Ni et al., 2022) | CVPR-22 | 0.779 | 0.743 | 0.603 | 0.802 | 0.732 |
| RECCE (Cao et al., 2022) | CVPR-22 | 0.767 | 0.731 | 0.609 | 0.812 | 0.730 |
| UCF (Yan et al., 2023) | ICCV-23 | 0.779 | 0.752 | 0.646 | 0.807 | 0.746 |
| F-G (Lin et al., 2024) | CVPR-24 | 0.744 | – | – | **0.848** | 0.796 |
| *CLIP-Based Methods* | | | | | | |
| CLIP (Ojha et al., 2023) | CVPR-23 | 0.743 | 0.750 | 0.730 | – | 0.741 |
| Wavelet-CLIP (Baru et al., 2024) | WACV-25 | 0.756 | 0.759 | 0.732 | – | 0.749 |
| **INFER (Ours)** | **–** | **0.863** | **0.819** | **0.747** | **0.848** | **0.819** |

Table 1: **AUC performance across cross-dataset evaluations**. The top section lists general SOTA methods, while the bottom focuses on CLIP-based approaches, including the proposed *INFER*.

## 4.2 How Well Does INFER Separate Classes?

To further assess the discriminative ability of *INFER*, we evaluated how well its learned representations separate classes in the feature space. We employed the Silhouette score as the clustering metric (↑ higher indicates better separation). Specifically, we compared *INFER* against two baselines, RGB+FFT and FFT, and report both the absolute Silhouette scores and the relative percentage improvements. As shown in table 2, *INFER* consistently achieves substantial gains across all datasets, highlighting its effectiveness in producing more separable feature clusters.

Table 2: Silhouette score (↑ higher is better) of INFER, RGB+FFT, and FFT.

| Dataset | INFER | RGB+FFT | FFT |
|---|---|---|---|
| CDF$_{v1}$ | 0.0617 | 0.0416 | 0.0393 |
| CDF$_{v2}$ | 0.0320 | 0.0168 | 0.0167 |
| FSh | 0.0217 | 0.0078 | 0.0072 |
| DFD | 0.0737 | 0.0249 | 0.0239 |

## 4.3 Ablation Studies

An ablation study was conducted to evaluate which modality provides the most discriminative information when combined with CLIP embeddings for the task of deepfake detection. The comparison

includes the standard CLIP module, as well as additional fusion configurations as described below. In the setting labeled FFT, the Fourier transform of the input image is processed through a shallow CNN and its embeddings are concatenated with CLIP features. In the RGB+FFT configuration, both RGB and FFT representations are passed through separate shallow CNNs, and their respective embeddings are fused with CLIP embeddings.

| Method | $CDF_{v1}$ | $CDF_{v2}$ | Avg. |
|--------|-----------|-----------|------|
| CLIP | 0.743 | 0.750 | 0.7465 |
| FFT | 0.759 | 0.760 | 0.7595 |
| RGB+FFT | 0.786 | 0.794 | 0.7900 |
| *INFER* | **0.863** | **0.819** | **0.8410** |

Table 3: **AUC scores and average performance across CDF datasets**.

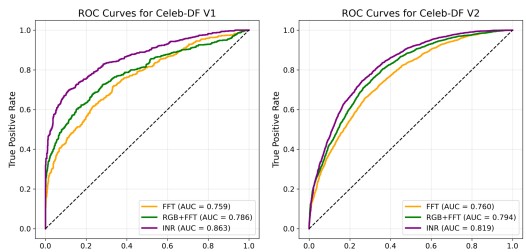

Figure 4: **ROC curves for $CDF_{v1}$ and $CDF_{v2}$**

As seen in Table 3, adding FFT features to CLIP embeddings improves the average AUC from 0.7465 to 0.7595 (**+1.71%**), close to Wavelet CLIP. Incorporating both RGB and FFT features further raises performance to 0.7900 (**+5.50%** over CLIP), confirming that spatial and spectral cues complement CLIP's semantics. Our INR-based method (*INFER*) achieves the highest performance with an average AUC of 0.8410, a **+6.06%** gain over RGB+FFT and **+11.24%** over CLIP. The corresponding ROC curves are shown in Figure 4. These results underscore the strong discriminative power of INR-derived features, which unify spatial–spectral information and expose subtle manipulation artifacts often missed by RGB or FFT features, supplying crucial cues that drive the performance gains of our approach.

## 5 CONFIGURATIONS AND ADDITIONAL PLOTS

The supplementary materials include detailed explanations of the network configurations used in the INR framework. These cover the selection of activation functions, the reasoning behind specific choices for network depth and the number of hidden neurons, as well as an analysis of why PCA provides better feature representations than $L_2$ norm-based maps. Moreover, additional visualizations are provided that demonstrate the INR's ability to capture multiscale structural information through its hierarchical decomposition. These materials offer further insight into the design choices and effectiveness of the proposed method.

## 6 CONCLUSION

In this work, we propose *INFER*, a deepfake detection framework that synergistically combines semantic embeddings from CLIP with spatial–spectral cues extracted from Implicit Neural Representations (INRs). Unlike traditional approaches that rely solely on either pixel or frequency-domain features, our method leverages INR-derived heatmaps, which capture multiscale structural patterns through a learned continuous implicit function. These heatmaps expose subtle inconsistencies often overlooked by CLIP and conventional CNN-based features. Through extensive experiments across standard deepfake detection benchmarks, we show that INR features significantly boost performance when fused with CLIP embeddings. Compared to standalone CLIP models, *INFER* achieves an average AUC improvement of **+11.24%**, and outperforms other CLIP-based variants such as RGB+FFT by **+6.06%**. These results underscore the complementary nature of INR-derived representations, which offer a richer and more discriminative feature space for detecting manipulated content. Our findings not only demonstrate the efficacy of INR-guided feature decomposition for deepfake detection but also open up new opportunities for applying INRs to other forensic tasks where subtle structural cues are critical. We believe this work lays the foundation for further exploration of implicit representations as a powerful modality in real-world multimedia integrity verification.

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

# A    SUPPLEMENTARY MATERIAL

## A.1    CHOOSING THE MOST EFFECTIVE ACTIVATION FUNCTION

As discussed in the main text, the core of an INR lies in its activation function. An inappropriate or conventional activation can often lead to degraded performance in image representation tasks. To assess the most effective activation function, we randomly sampled 100 real and 100 fake images from the FaceForensics++ dataset, following the preprocessing steps outlined in Section 3.1. INRs were then trained using sinusoidal activations from SIREN (Sitzmann et al., 2020), as well as those introduced in Gauss (Ramasinghe & Lucey, 2022) and WIRE (Saragadam et al., 2023).

The table below summarizes the average Peak Signal-to-Noise Ratio (PSNR, in dB) obtained for both real and fake images across the different activation types:

| Activation Function | PSNR (Real) | PSNR (Fake) |
|---|---|---|
| SIREN | 37.41 | 38.18 |
| Gauss | 29.41 | 29.71 |
| WIRE | 20.01 | 19.73 |

Table 4: Average PSNR values for real and fake images across different activation functions.

As shown in Table 4, the SIREN model with sinusoidal activation significantly outperforms both Gauss and WIRE across real and fake image reconstructions. Due to its superior performance, SIREN was adopted as the default activation function for all INR-based experiments in this work.

## A.2    CHOOSING THE NUMBER OF HIDDEN NEURONS

Another important design choice in INRs is the number of hidden neurons in each layer. Increasing this number generally enhances the network's representation capacity, enabling it to capture more complex structures and finer details. However, beyond a certain point, increasing the hidden neuron count may no longer lead to meaningful improvements in reconstruction quality. Specifically, the PSNR often plateaus once the network has reached its capacity to represent the target signal, indicating diminishing returns with further increases in model size. It is worth noting that this behavior can also depend on the type of activation function used.

Similar to the procedure described in Section A.1, we randomly sampled 100 real and 100 fake images from the FaceForensics++ dataset and varied the hidden neuron count from 32 to 160 in increments of 32 while keeping the number of hidden layers as two. The resulting average PSNR values for both real and fake images are presented in the left side of Fig. 5.

## A.3    CHOOSING THE NUMBER OF HIDDEN LAYERS

In addition to the number of hidden neurons, the depth of the network, defined by the number of hidden layers, is another key factor that influences the expressiveness of INRs. Deeper networks are generally capable of modeling more intricate patterns and hierarchical structures, potentially leading to better reconstruction quality. However, similar to increasing the number of neurons, increasing the number of hidden layers may also yield no further improvements in reconstruction quality. This phenomenon can be attributed to the combined effects of the activation function and other network parameters.

To analyze the impact of network depth, we varied the number of hidden layers from 1 to 3 while keeping the number of hidden neuron count as 128. Following the same evaluation protocol as before, we randomly sampled 100 real and 100 fake images from the FaceForensics++ dataset and trained INRs under each configuration. The average PSNR values obtained for both real and fake images are summarized in the right side of Fig. 5.

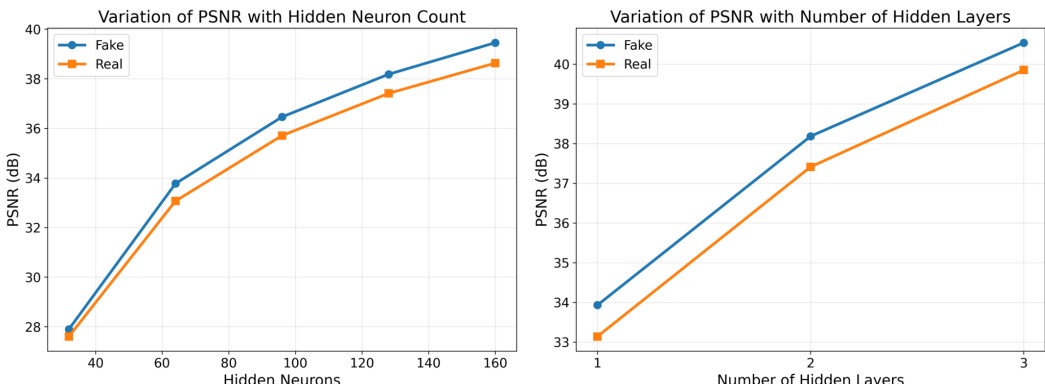

Figure 5: **Average PSNR Variation for both Real and Fake Samples:** Left side plot shows how the average PSNR varies with hidden neuron count while the Right side plot shows how the average PSNR varies with the number of hidden layers

### A.4 UTILIZED INR

For the image reconstruction task through INR, our objective is to achieve at least 35 dB PSNR, as this level reflects high signal fidelity and indicates that the INR has effectively captured the essential structural content of the image. Such a threshold helps ensure that the reconstruction is stable and reliable for downstream analysis, including feature extraction and classification. At the same time, we aimed to avoid overly complex networks with a large number of trainable parameters. To balance reconstruction quality and model efficiency, we selected an INR architecture with sinusoidal activation sitzmann2020implicit, consisting of 128 hidden neurons and 2 hidden layers.

### A.5 INR RECONSTRUCTIONS

In addition to proving the quantitative results for INR reconstruction, Figure 6, and Figure 7 showcase how the INR reconstruction quality looks for six different real and fake samples respectively.

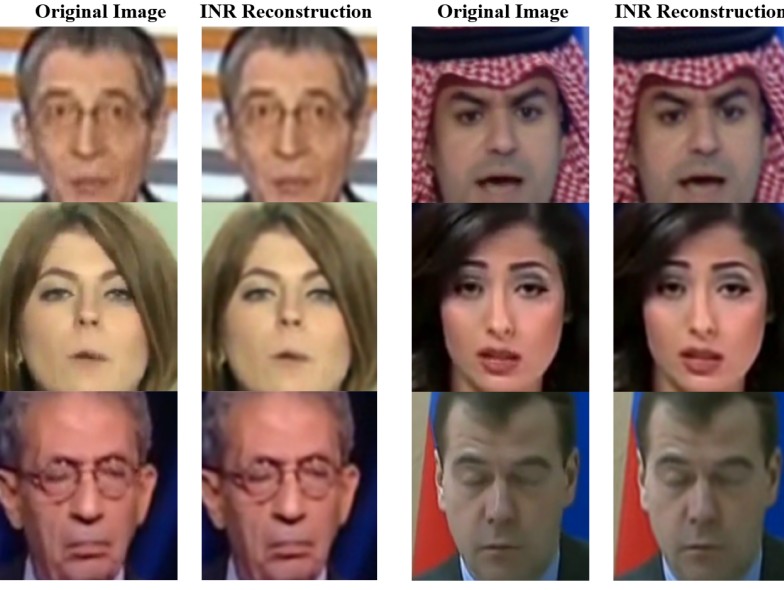

Figure 6: **Original Images and INR Reconstructions for Real Samples**: This figure presents side-by-side comparisons of original real images and their corresponding reconstructions produced by INRs.

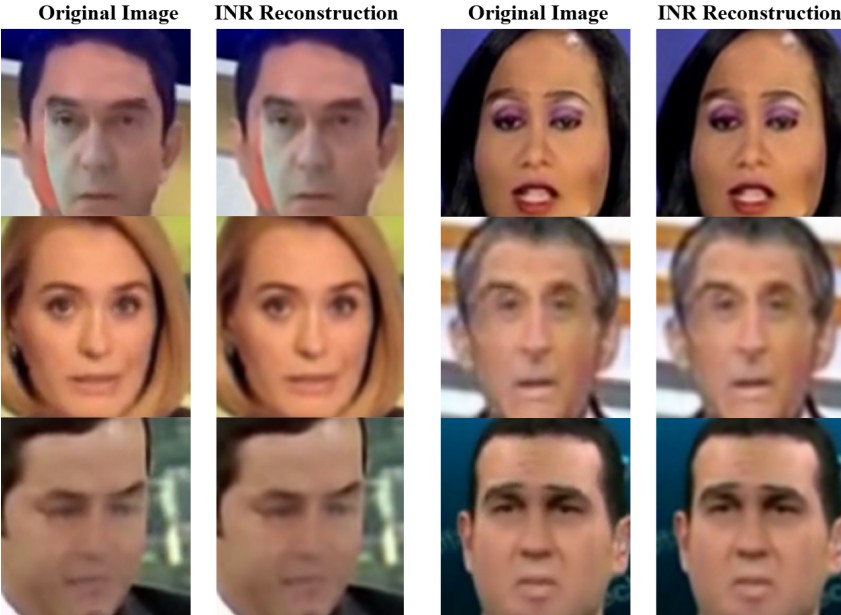

Figure 7: **Original Images and INR Reconstructions for Fake Samples**: Side-by-side comparisons of original fake images and their corresponding INR reconstructions.

## A.6 HEATMAP ANALYSIS FOR DIFFERENT DATASETS

In addition to the heatmap visualizations from the $CDF_{v2}$ dataset in the main text, we also present INR-derived heatmaps for $CDF_{v1}$, DFD, and FSh. These additional visualizations further highlight the ability of INRs to capture structural inconsistencies across different manipulation methods and datasets.

### A.6.1 $CDF_{v1}$

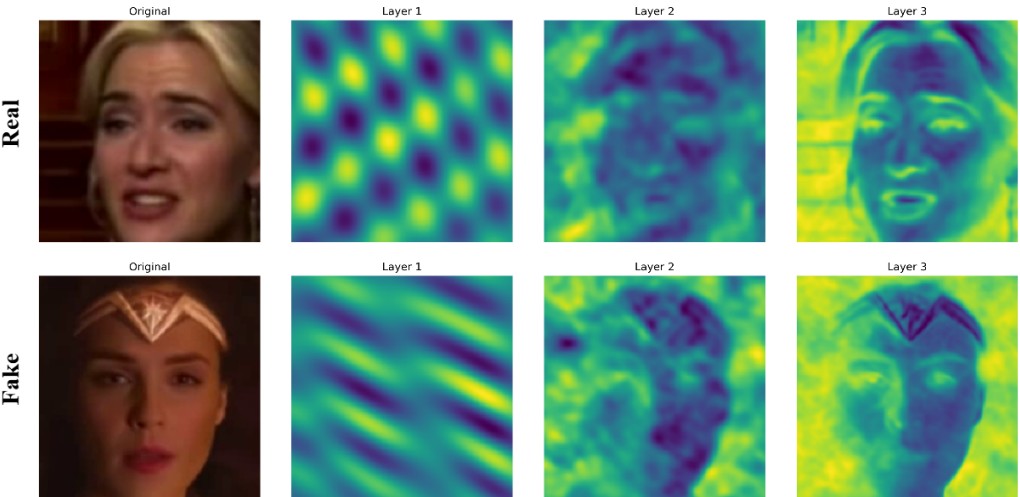

Figure 8: **INR Feature Heatmap Progression for Real and Fake Images ($CDF_{v1}$)**

As can be seen from Figure 8, the first row corresponds to a real image, while the second row shows a deepfake. In the real image, the INR learns progressively meaningful representations: the first layer

captures periodic frequency patterns, the second begins to reveal coarse facial structure, and the third cleanly delineates key semantic features such as eyes, nose, and mouth with sharp transitions and spatial coherence. This reflects a natural multiscale decomposition that can be commonly observed in INRs trained on natural content. In contrast, the heatmaps from the deepfake image reveal subtle inconsistencies. While the initial layer shows strong frequency bands, the second and third layers display noisier, less structured activations, particularly in regions like the cheek and jawline. Notably, the third-layer features lack the same spatial sharpness and exhibit localized overactivation near synthetic textures (e.g., the forehead accessory). These differences highlight how INR activations implicitly encode artifacts introduced by manipulation, supporting their utility in forensic analysis.

### A.6.2 DFD

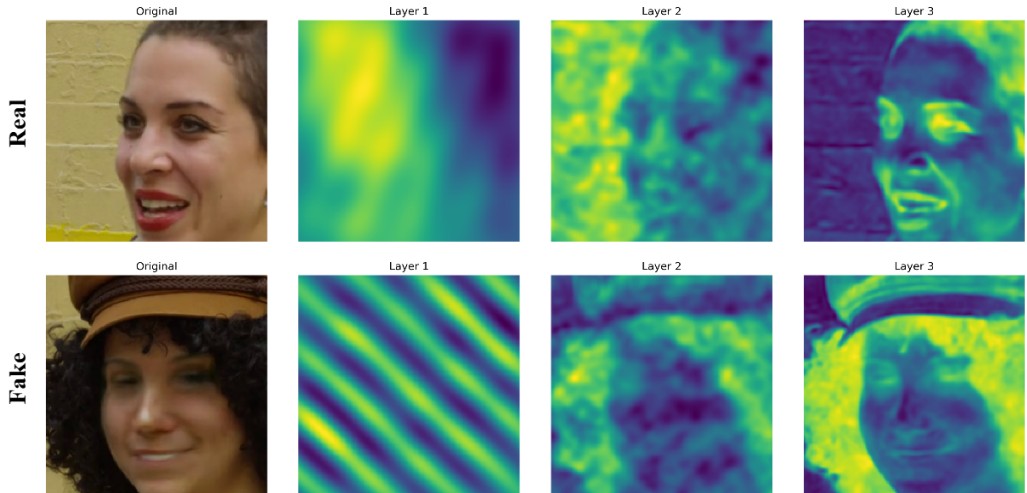

Figure 9: **INR Feature Heatmap Progression for Real and Fake Images (DFD)**

As can be seen from Figure 9, in the real sample (top row), the network exhibits a natural decomposition: the first layer encodes smooth, low-frequency gradients, while subsequent layers progressively extract spatial structure aligned with facial semantics. By the third layer, the representation distinctly highlights the subject's facial features and background texture in a spatially coherent manner. On the other hand, the fake sample reveals signatures of overactivation and structural inconsistency. As the depth increases, the heatmaps become increasingly noisy, with attention distributed unevenly across irrelevant regions such as the background or accessories (e.g., hat, hair). The third layer lacks the focused delineation observed in the real case, underscoring the INR's struggle to generalize to synthetic artifacts. These observations highlight the discriminative potential of INR-derived features in distinguishing real from fake content.

### A.6.3 FSH

As can be seen from Figure 10, in the real image (top row), the network exhibits a natural and structured activation flow. The first layer encodes smooth, diagonal sinusoidal frequencies. By the second layer, coherent facial structures begin to emerge. In the third layer, semantic features such as the eyes, mouth, and hairline become sharply defined, with strong localization and contrast — indicating confident learning of meaningful spatial content. In contrast, for the fake image, the deep layers tend to be spatially noisy and less well-formed activations in layers 2 and 3. Although the overall face layout is still present, the details are less distinct. Key features like the mouth and eyes appear blurred or over-smoothed, and the network spreads attention more uniformly, suggesting difficulty in modeling fine-grained semantics. These differences align with patterns observed across fake content, where subtle inconsistencies in structure and texture impede robust INR representation learning. This highlights the sensitivity of INR-derived heatmaps to manipulation artifacts.

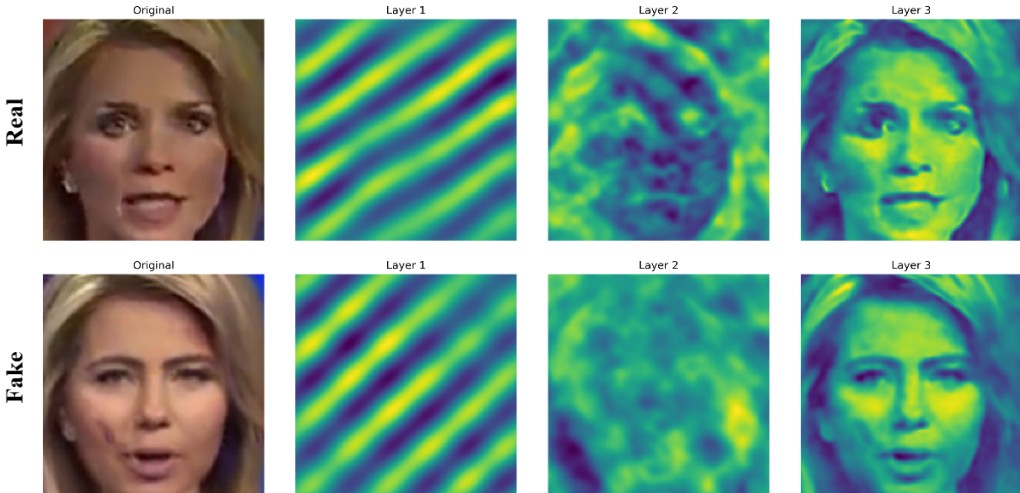

Figure 10: **INR Feature Heatmap Progression for Real and Fake Images (FSh)**

### A.7 FEATURE SPACE ANALYSIS

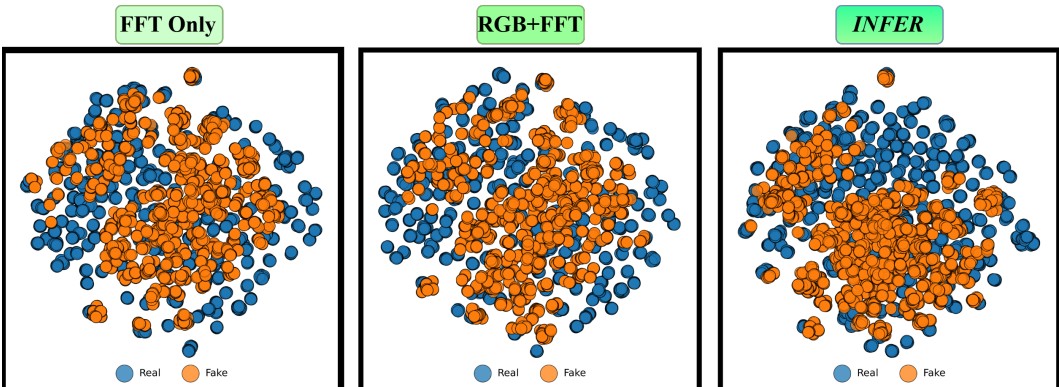

Figure 11: **t-SNE visualization of feature embeddings from the CDF$_{v2}$ dataset using different input modalities**

To better understand how different feature combinations affect the structure of the learned representation space, we visualize the embeddings of real and fake samples using t-SNE for three configurations as shown in Figure 11. Each configuration involves concatenating the respective features before classification. These plots reveal how the choice of representation transforms the feature space and impacts class separability.

**FFT Only (Left):** This configuration concatenates global frequency information (via the FFT magnitude spectrum) with CLIP embeddings. The FFT captures the global energy distribution across frequencies, but discards all spatial localization. While this can detect abnormal high-frequency content typical of manipulations, it cannot tell where these signals occur which is a critical limitation for identifying local artifacts. As many fake traces are spatially sparse or structured (e.g., boundary mismatches or warped facial regions), this global representation leads to significant overlap between real and fake distributions in the t-SNE space. Moreover, FFT is phase-agnostic in this setup, meaning structural information embedded in phase is ignored. CLIP contributes semantic context but lacks pixel-level sensitivity. As a result, the combined representation fails to disentangle class boundaries effectively.

**RGB + FFT (Middle):** Here, raw image pixels, FFT features, and CLIP embeddings are concatenated. While this introduces spatial information through RGB and captures frequency cues through FFT,

the representation is not explicitly organized to reflect multi-scale spatial-frequency patterns. Even though FFT complements this with frequency statistics, it still lacks localization. Consequently, the feature space becomes more structured than the FFT-only case, but real and fake samples still exhibit considerable intermixing, suggesting insufficient separation.

**INFER (Right):** The proposed *INFER*, where INR-derived heatmaps are concatenated with CLIP embeddings, results in the most well-separated clusters. INRs reconstruct images from continuous coordinates, and the resulting heatmaps capture how different spatial positions activate the network. These activations inherently encode localized frequency responses, much like a learned multiscale basis decomposition. From a signal processing perspective, INRs offer a unique advantage: they disentangle an image's representation into a hierarchy of frequencies conditioned on position. This means they capture both what frequencies are present and where, which is similar to a spatially adaptive filter bank. Fake images, which often contain unnatural local discontinuities, exhibit distinct activation behaviors in these heatmaps compared to real images. When concatenated with CLIP, which provides semantic structure, the combined representation becomes highly expressive: local inconsistencies are aligned with global semantics, resulting in a well-structured, and a more separable space. This is visually evident from the transformation that both real and fake clusters have undergone compared to Left and Middle figures.

### A.8 EFFECT OF NUMBER OF HIDDEN LAYERS OF INR IN DEEPFAKE DETECTION

To examine the role of network depth in our INR-based DeepFake detection framework, we conducted an ablation study by progressively incorporating feature maps from different hidden layers. We first considered only the feature map from the first hidden layer, then combined feature maps from the first and second layers, and finally aggregated feature maps from all three layers. This step-by-step inclusion allows us to assess how deeper representations contribute to the discriminative power of the model. The resulting AUC scores across the test datasets are summarized in Table 5

Table 5: AUC scores with increasing number of INR layer features across datasets.

| Feature Combination | Celeb-DF v1 | Celeb-DF v2 | Fsh | DFD |
|---|---|---|---|---|
| Single Layer Feature | 0.8258 | 0.7916 | 0.7413 | 0.8399 |
| Two Layer Features | 0.8341 | 0.8071 | 0.7430 | 0.8369 |
| Three Layer Features | **0.8630** | **0.8190** | **0.7470** | **0.8480** |

### A.9 RELATIVE CONTRIBUTION FROM CLIP

To asses how INR alone helps in deepfake detection task, we conducted an ablation study in which the classifier was trained using only PCA-based INR features. The results, shown in Table 6 , compare this INR-only baseline to INFER:

Table 6: How INR only features help for deepfake detection

| Method | Celeb-DF v1 | Celeb-DF v2 | FSh | DFD |
|---|---|---|---|---|
| INR-only | 0.551 | 0.580 | 0.543 | 0.506 |
| **INFER** | **0.863** | **0.819** | **0.747** | **0.848** |

These results highlight the importance of CLIP features within the INFER framework. However, it is crucial to emphasize that the strength of INFER lies in the complementary nature of these two modalities. INR-derived heatmaps provide a rich spatial–frequency decomposition, capturing structural irregularities and local manipulation artifacts that may be imperceptible in RGB space. CLIP, on the other hand, contributes global semantic understanding—such as identity coherence and contextual realism—that helps place these local distortions in a broader, meaningful context.

## A.10 GRAD-CAM ANALYSIS

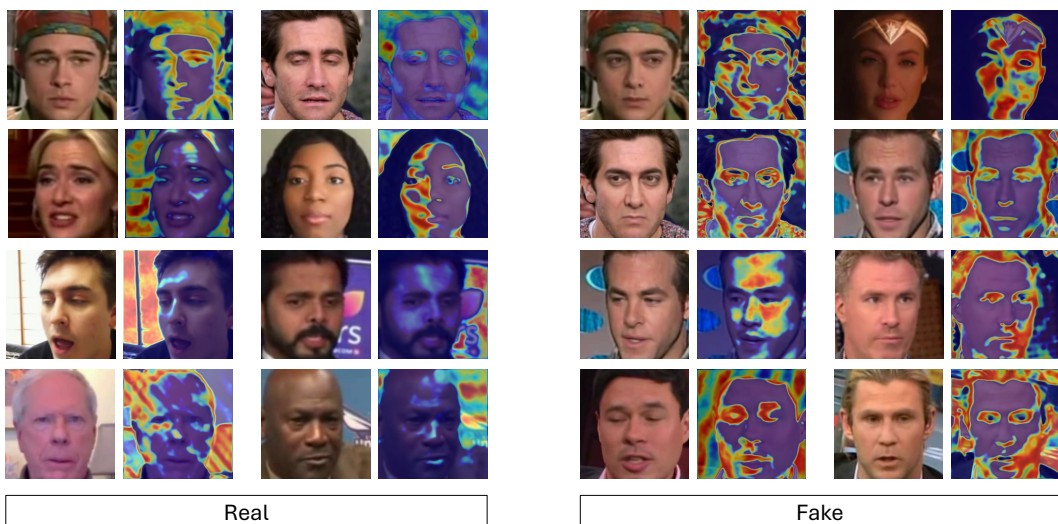

Figure 12: **Activation Maps for Real and Fake Images**

The Grad-CAM visualizations reveal distinct attention patterns for real and fake images, highlighting the complementary roles of semantic and structural cues in *INFER*, as shown in Figure 12. For real faces, the heatmaps are diffuse, with activations spilling into the background and being distributed across broad facial regions rather than tightly clustering around specific landmarks. This suggests that, in the absence of obvious distortions, the detector relies on the overall consistency of textures, both in the background and on the face, rather than on narrowly defined semantic features. In contrast, when processing deepfake images, the attention drifts outward toward peripheral zones such as the hairline boundaries and jawline contours, as well as toward landmark regions like the eyes, nose, and mouth. These are precisely the areas where synthesis artifacts commonly appear, including blending errors, texture irregularities, and subtle warping. This shift in attention arises from *INFER*'s integration of INR-derived features: by overfitting a sinusoidally activated INR to each input and extracting multiscale activation heatmaps via PCA, *INFER* captures fine-grained frequency-domain distortions that standard CNN backbones and CLIP embeddings often overlook. When these INR heatmaps are concatenated with CLIP's semantic embeddings, the downstream classifier learns to look where the fakes break, prompting Grad-CAM to highlight artifact-rich regions in fake images. Consequently, *INFER* enhances robustness by guiding the detector to attend not only to plausible facial geometry but also to the subtle structural inconsistencies that are characteristic of deepfakes.

## A.11 ROBUSTNESS ANALYSIS UNDER DEGRADATOPMS

As an addtional ablation study, We evaluated the performance of three methods—INFER (ours), RGB+FFT, and FFT-only—under three common perturbations: Gaussian blur ($\sigma = 1$, kernel size $5 \times 5$), JPEG compression (Q=30), and additive Gaussian noise (std=10). Experiments were conducted on both Celeb-DF v1 and v2, and we report the AUC before and after degradation, and the resuls are shown in Table 7

Across both datasets, **INFER consistently demonstrates the smallest performance drop under all perturbations**. For example:

- **Blur:** INFER drops only **4.06%** on Celeb-DF v1 and **3.91%** on v2, compared to maximum drops of **11.05%** (FFT) and **6.67%** (RGB+FFT), respectively.

- **Noise:** INFER drops **7.30%** (v1) and **4.03%** (v2), while FFT suffers the most with **15.13%** and **5.66%**.

- **Compression:** Overall, all methods tend to remain stable, but INFER again shows the lowest drop (**0.46%** on v1, **3.17%** on v2).

Table 7: AUC before and after degradation on Celeb-DF v1 and v2.

| Dataset | Method | Blur | | Compression (Q30) | | Noise (std=10) | |
|---|---|---|---|---|---|---|---|
| | | Initial | After | Initial | After | Initial | After |
| Celeb-DF v1 | INFER | 0.863 | **0.828** | 0.863 | **0.859** | 0.863 | **0.800** |
| | RGB+FFT | 0.786 | 0.712 | 0.786 | 0.775 | 0.786 | 0.687 |
| | FFT | 0.759 | 0.675 | 0.759 | 0.737 | 0.759 | 0.644 |
| Celeb-DF v2 | INFER | 0.819 | **0.787** | 0.819 | **0.793** | 0.819 | **0.786** |
| | RGB+FFT | 0.794 | 0.741 | 0.794 | 0.774 | 0.794 | 0.746 |
| | FFT | 0.760 | 0.721 | 0.760 | 0.738 | 0.760 | 0.717 |

These results highlight the robustness of our INR-based representations, which are more resilient to pixel-level corruption than image- or frequency-based baselines.

A.12 LLM USAGE

LLMs were used to improve the writing of the paper.

