# OpenReview forum: "INFER: Implicit Neural Features for Exposing Realism"
_ICLR.cc/2026/Conference — ICLR 2026 Conference Withdrawn Submission_

### Official Review · Reviewer_ny8w · 2025-10-24

**Soundness:** 2
**Presentation:** 2
**Contribution:** 2
**Rating:** 2
**Confidence:** 4

**Summary:**

This paper proposes using multi-layer spatial–spectral heatmaps from INR internal activations compressed via PCA as forensic features, combined with CLIP visual representations for real/fake classification. INR’s sensitivity to continuity and frequency reveals multi-scale structural differences that complement CLIP’s global semantics, improving separability and robustness. To adapt to smooth, low-frequency-redundant signals, the authors design a compact convolutional heatmap encoder and train the system end-to-end for classification. The goal is to improve deepfake detection accuracy and provide interpretable cues. Extensive experiments on standard deepfake benchmarks show clear superiority over existing methods, highlighting the method’s generalization, robustness, and practicality.

**Strengths:**

(1) Successfully uses INR internal activations compressed via PCA into multi-layer spatial and spectral heatmaps as forensic features, combined with CLIP visual representations for classification; experiments show significant performance gains after fusion, demonstrating effectiveness for deepfake detection.
(2) The idea is insightful: INR activations enhance class separability, point out limitations of traditional RGB/FFT representations, and show that changing the feature space can improve separability.
(3) The effectiveness of the method has been proven through a series of experiments.

**Weaknesses:**

(1) the overall novelty is somewhat insufficient; the combination of existing components and the improvements need clearer delineation of what is genuinely new. (2) The fusion mechanism is not clearly explained, lacking visualization and quantitative fusion details. (3) For the challenge of unseen operations and cross-dataset generalization, the theoretical motivation is strong but experimental coverage is limited, failing to adequately demonstrate generalization and robustness. (4) Although the framework is simple, per-image INR fitting may incur substantial compute and time costs and may rely on dataset-specific artifacts, raising concerns about deployability and cross-domain robustness; the paper lacks a thorough cost report and significance comparisons, so the practical effectiveness of the core contribution in real scenarios requires stronger evidence.

**Questions:**

(1) Figure 1 is not visually appealing.
(2) The specific role of the CLIP encoder and its contribution path are not clearly described; please clarify.
(3) Please provide the detailed construction process of the INR heatmaps and add more steps in Fig. 1 (activation extraction levels, PCA (4) dimension and number of components, alignment strategy, etc.).
(4) Please more clearly quantify the complexity and time cost of INR fitting and explain possible acceleration schemes and performance trade-offs.
(5) The font in Table 4 is small; please increase the font size for readability.
(6) In Appendix A6, please provide more visual examples for each dataset to enable better comparison and analysis.

---

### Official Review · Reviewer_HwuA · 2025-10-31

**Soundness:** 3
**Presentation:** 2
**Contribution:** 3
**Rating:** 4
**Confidence:** 3

**Summary:**

This article proposes a deepfake detection framework in the context of face images from videos. The designed method relies on training a classifier with two distinct inputs: CLIP embeddings and an analysis of Implicit Neural Representation (INR) features. Performance is compared on several out-of-distribution datasets against different ablations of the method, as well as a set of common baseline techniques.

**Strengths:**

The article provides a thorough analysis of the benefits of using the INR, with an in-depth examination of the features produced and their significance. The design choices are clearly stated and explained, with complete ablation studies and comparisons where necessary. The reported experimental results show substantial performance gains, proposing an adequate solution to the considered problem.

**Weaknesses:**

The authors adopt a broad definition of deepfakes, encompassing all artificial content designed to mimic real data, and present their tool as a deepfake detection framework. However, the scope they consider (face images extracted from videos) is significantly narrower. It should be clearly stated to which modalities the proposed method is intended to apply.

The datasets considered for evaluation are outdated and do not represent state-of-the-art deepfake generation. Notably, most of the evaluated datasets only consider face swapping/reenactment, and the only one that does not (DeepFakeDetection) is similar to the training set. Including an evaluation on a more recent dataset, such as DF40 [1], would greatly benefit this article. This would allow for a more meaningful comparison of the proposed method’s effectiveness relative to current advances in deepfake detection.

We regret the authors did not provide the code used for their experiments nor expressed their intention to do so. This seriously undermines the reproducibility of this work and the trustworthiness of the results reported.

Minors:
- The figures are rendered as pixelized images, decreasing the overall quality and text readability
- Figure 2: usage of the acronym “t-SNE” without explanation in the caption
- l.215 grammar error “demonstrate on how”
- l.317 reference to “Section 3.4.5” instead of Figure 3
- l.835 bad citation “sitzmann2020implicit”
- l.1119 typo in subsection title “Degradatomps”

[1] Yan, Z., Yao, T., Chen, S., Zhao, Y., Fu, X., Zhu, J., Luo, D., Yuan, L., Wang, C., Ding, S., et al. (2024). DF40: Toward next-generation deepfake detection.

**Questions:**

The training cost of an INR can be significant, taking up to several minutes per image. Could the authors comment on the computational cost of their method, particularly compared to those of their baselines?

The article disregards some of the state-of-the-art methods arguing they cannot compare with methods using data augmentation. Could the authors clarify why methods using data augmentation should be considered separately? What is the benefit from not requiring data augmentation during the training? Could the authors present the results of their method when data augmentation is used?

Additionally, some of the baselines considered do use data augmentation during their training phase (e.g. the UCF article clearly states “We also apply some widely used data augmentations” [2]). Why are these not disregarded while other works are?

Similar to previous works, the authors consider for their classifier input the CLIP embeddings of the evaluated images. However, CLIP comes with the benefit of mapping both image and text to the same embedding space, which is not used here. Why didn’t the authors consider other image foundation models (e.g., DINOv3 [3])?

The authors evaluate their method against diverse degradations of the inputs, but do not compare the performance loss with that of baseline methods. Could this be reported as well?

[2] Z. Yan, Y. Zhang, Y. Fan and B. Wu, UCF: Uncovering Common Features for Generalizable Deepfake Detection, 2023 IEEE/CVF International Conference on Computer Vision (ICCV), page 6

[3] Siméoni, O., Vo, H. V., Seitzer, M., Baldassarre, F., Oquab, M., Jose, C., ... & Bojanowski, P. (2025). DINOv3. arXiv preprint arXiv:2508.10104.

---

### Official Review · Reviewer_qxgY · 2025-11-01

**Soundness:** 2
**Presentation:** 3
**Contribution:** 3
**Rating:** 4
**Confidence:** 4

**Summary:**

The authors propose to use a CLIP encoder together with feature maps of an implicit neural network for the detection of GAN-generated images.

**Strengths:**

- Clear idea
- several ablation studies are present

**Weaknesses:**

two clarity issues:

- 1 Details to the training of the INR are largely absent. Given that this is the one component that makes the difference, and a less common one, this is a problem for reproduction of results.
Is the INR fitted to a single image as one would expect ? It is trained until what ? Fixed amount of iterations or some other criterion ?



- 2 details for the PCA are missing: does one fit one PC component for all positions in the image ? does one get one PC position for each position in the image separately ? the PCA is fitted over what set ? all features over all positions for a single image ? or over images from some set?

- the datasets are exclusively from the era before diffusion models. This could mean that it might possibly generalize differently to diffusion based datasets.

- an analysis is missing why INR features are different for real vs GAN-sourced images. INR is a reconstruction approach. Why it fails on GAN generated images ? or  ... is it spectral energy in some bands as Table 5 and the choice of sinusoids as activation suggest ?  In that context, one can cite Chandrasegharan, ECCV 2022 which identified color distribution for similar early datasets.

The paper is not bad at all. It can get a higher rating when one would try on diffusion datasets, and by clarifying INR training, also by trying to get an explanation what of the INR features make a difference.

**Questions:**

See questions about INR training and PCA details in the weaknesses section.

- INR training protocol
- PCA estimation and application details
- maybe a newer dataset
- an analysis why INR makes a difference. Since it is a reconstruction approach, it should be equally ok to reconstruct fake images ??

I understand that INR is a shallow small MLP type network, but anyway, how long does it take to train it per image ?

---

### Official Review · Reviewer_42C3 · 2025-11-02

**Soundness:** 2
**Presentation:** 3
**Contribution:** 2
**Rating:** 4
**Confidence:** 3

**Summary:**

This paper proposed a framework called INFER, which leverages Implicit Neural Representations (INRs) and CLIP for deepfake detection. The proposed method outperforms previous ones including general and CLIP-based approaches on 4 widely-used OOD deepfake detection benchmarks (CDFv1,CDFv2,FSh,DFD)

**Strengths:**

Effectively explore spatial-spectral decomposition capability of INRs and improve upon previous works, especially with CLIP-based ones.

The proposed method achieves good performance on 4 OOD deepfake detection datasets for both approaches such as normal and CLIP-based.

**Weaknesses:**

It needs more focus on explaining how INRs improve the deepfake performance. The authors mentioned INRs were used in different tasks but none applies it for deepfake detection. The experimental design is not really clear what advantages of INRs really help to distinguish real and fake images and then improve the performance of deepfake detection.

The authors claimed this paper is the first work using INRs for deepfake detection, but there is another paper which also uses INRs for deepfake detection (https://link.springer.com/chapter/10.1007/978-3-031-99565-1_18). The authors need to compare with this work.

**Questions:**

Following [1], it explored the frequency information from the features of CLIP. The proposed paper designed a parallel branch and used the INRs to leverage frequency information then concatenate with CLIP features for the classification head. The question here is where does the efficiency come from that will bring a better improvement, is it better to explore the frequency from CLIP features or directly from the frequency input or leverage both?

It would be good to compare frequency-based approaches with INRs or INRs are the most suitable one for deepfake detection?

FFT and RGB are baselines but they are a part of the proposed INFER network or they are separately just an ablation for INFER. Since they are passed through separate shallow CNNs then how do they relate to the INFER network and the improvement of INFER?

How about the performance if using CLIP with RGB or FFT (which passes through a shallow CNN). If they are available in table 2 & 3, the author can adjust them to make it clear.

---

### Note · Authors · 2025-11-13

I have read and agree with the venue's withdrawal policy on behalf of myself and my co-authors.